# Fungal Aeroallergens—The Impact of Climate Change

**DOI:** 10.3390/jof9050544

**Published:** 2023-05-07

**Authors:** Monika Sztandera-Tymoczek, Agnieszka Szuster-Ciesielska

**Affiliations:** Department of Virology and Immunology, Institute of Biological Sciences, Maria Curie-Skłodowska University, Akademicka 19, 20-033 Lublin, Poland; monika.sztandera-tymoczek@mail.umcs.pl

**Keywords:** overreactivity-related diseases, fungal-related allergic diseases, microfungi, climate change

## Abstract

The incidence of allergic diseases worldwide is rapidly increasing, making allergies a modern pandemic. This article intends to review published reports addressing the role of fungi as causative agents in the development of various overreactivity-related diseases, mainly affecting the respiratory tract. After presenting the basic information on the mechanisms of allergic reactions, we describe the impact of fungal allergens on the development of the allergic diseases. Human activity and climate change have an impact on the spread of fungi and their plant hosts. Particular attention should be paid to microfungi, i.e., plant parasites that may be an underestimated source of new allergens.

## 1. Introduction

Allergies are a group of complex diseases characterized by an inappropriate or overreactive immune response of the organism to common environmental substances named allergens. Most allergens, e.g., house dust mites, pollen, fungal or mold spores, food (especially milk, eggs, fruit, wheat, soybeans, seafood, and nuts), insect venoms (e.g., wasps and bees), some medications, latex, and household chemicals, are harmless to humans. Upon contact with these agents, a hypersensitivity reaction develops and antibodies are produced in predisposed subjects [1,2]. Depending on the cause, i.e., the type of contact with the allergen, the following conditions are distinguished: (1) respiratory system allergies, i.e., allergic rhinitis with conjunctivitis and asthma with wheezing, coughing, and shortness of breath; (2) contact (skin) allergies, among which the most common are atopic and contact dermatitis, manifested mainly in rashes; (3) food allergies, and (4) insect venom allergies, which cause a wide range of symptoms and can be life-threatening in some cases (anaphylaxis) [3,4,5].

## 2. Allergy Epidemiology

The incidence of allergic diseases in the world is rapidly increasing, and according to the estimates of the World Allergy Organization (WAO), it ranges from 10–40%, depending on the country. In most developed countries, allergy affects over 20% of the population [6]. While allergy was regarded as a rare disease in the early 20th century, the last few decades have witnessed a dramatic increase in its prevalence. Currently, more than 150 million Europeans suffer from chronic allergic diseases, and 20% struggle with severe and debilitating forms. By 2025, half the European Union population is estimated to suffer from allergies. The prevalence of allergic diseases is increasing together with the progress in urbanization, industrialization, pollution, and climate change, factors that are not likely to change quickly. Moreover, these data are considered to be underestimations, because many patients do not report their symptoms or are misdiagnosed, with potentially 45% of patients never receiving an allergy diagnosis [7].

Respiratory allergies are the most common allergy type in Europe and worldwide [8,9]. Data in the White Book of Allergy published in 2013 by the WAO confirm that the prevalence of allergic rhinitis and asthma is increasing worldwide [10]. Allergic rhinitis with conjunctivitis is the most common non-infectious rhinitis, affecting approximately 400 million people worldwide [11,12]. Asthma is one of the most common chronic diseases, affecting about 339 million people worldwide, and its prevalence is increasing, especially among children [13,14]. The most common asthma phenotype is allergic asthma. It is estimated that up to 89% of childhood asthma and more than 50% of adult asthma cases may have an allergic component [15]. According to the European Federation of Allergy and Airways Diseases Patients’ Associations (EFA), allergic rhinitis affects 4–40% of Europe’s population [16]. In comparison, the prevalence of asthma in the European Union is 8.2% in adults and 9.4% in children [17]. In 2019, Portugal and Sweden were the EU countries with the highest prevalence of asthma, with approximately 9.9% and 8.2% of their populations suffering from the disease, respectively [18]. 

## 3. Types of Hypersensitivity Reactions

One of the functions of the human immune system is to provide protection against such microorganisms as bacteria, viruses, and fungi. Exaggerated immune responses and hypersensitivity reactions may lead to disease development. In 1963, Gell and Combs, who investigated the effector mechanism responsible for cell and tissue damage and the type of immune response, created the basis for the classification of hypersensitivity reactions, dividing them into four distinct types: type I (immediate or IgE-dependent), type II (cytotoxic or dependent on IgG/IgM), type III (immune complexes), and type IV (delayed or dependent on T lymphocytes) [19]. However, in clinical practice, the categories of hypersensitivity may overlap, because patients exhibit simultaneous coexistence of symptoms characteristic of specific hypersensitivity reactions [20,21].

In type I hypersensitivity, type 2 helper T cells (Th2) and their mediators initiate a change in the isotype of B cells, producing allergen-specific IgE. IgE antibodies bind to receptors (FcεRI) present on the surface of mast cells and basophils [21]. Upon re-exposure, the allergen is bound by IgE, resulting in an immediate type I hypersensitivity reaction in two phases. The early phase occurs within minutes of allergen exposure and is triggered by histamine, proteases, lysosomal enzymes, and other mediators released after the degranulation of mast cells and basophils. In addition, mast cells produce lipid mediators, including prostaglandin D2 and leukotriene D4, from arachidonic acid, and release them into the circulation. The late phase begins 4 to 8 h after allergen exposure and is mediated by cytokines IL-1, IL-4, IL-5, IL-13, tumor necrosis factor (TNF-α), and granulocyte-monocyte colony-stimulating factor (GM-CSF) [21,22,23] (Figure 1). Immediate hypersensitivity reactions include anaphylaxis, bronchial asthma, and urticaria [21,24]. 

Unfortunately, antibodies can sometimes bind to self-antigens, directing a cytotoxic response against the host. This is the basis of type II hypersensitivity, a cytotoxic reaction characterized by IgG/IgM antibodies recognizing self-antigens usually found on circulating blood cells (erythrocytes, neutrophils, platelets), epithelial cells of glands and mucous membranes, or basement membranes [21]. Type II reactions are divided into two subtypes: IIa and IIb. Type IIa refers to reactions characterized by cytolytic destruction of target cells. IgG/IgM binding on the cell surface triggers cytotoxicity via three mechanisms [25]. In the first one (antibody-dependent cell cytotoxicity, ADCC), IgG binds to the Fc gamma IIb receptor (FcγRIIb) on NK cells and macrophages, which leads to their degranulation and release of perforin and granzyme directly destroying the cells. In the second mechanism (complement-dependent cytotoxicity, CDC), antigen-antibody complexes on the cell surface activate the complement via the classical pathway to form the C5–C9 membrane attack complex (MAC), which induces lysis of the target cell. The third mechanism (antibody-dependent cellular phagocytosis, ADCP) initiates phagocytosis by binding IgG and IgM to Fc receptors on phagocytes and activating them (Figure 2). Type IIb reactions refer to the direct stimulation of cells by autoantibodies and the development of disease; for example, in Graves’ disease, antibodies directed against thyrotropin receptors stimulate the thyroid gland to produce excessive amounts of the hormone [27,28]. 

In the type III immune response, class G and M immunoglobulins bind to antigens, forming immune complexes that, deposited in tissues, activate complement system components. This leads to the recruitment and activation of neutrophils, mast cells, and basophils, which cause inflammation and tissue damage [21,25] (Figure 3). The type and nature of symptoms occurring in type III reactions depend on the site of deposition of immune complexes rather than on the source of the allergen [27,31]. The antigens involved in the type III response may be identified as self, as in the case of autoimmune diseases (e.g., lupus), or non-self [27].

Type IV hypersensitivity reactions are referred to as delayed reactions. The primary effector cells are T lymphocytes that can cause injury directly (cytotoxic T lymphocytes, Tc) or activate other leukocytes (macrophages, neutrophils, and eosinophils) to damage tissues by releasing reactive oxygen species, lysosomal enzymes, and inflammatory cytokines (helper T cells) [33]. The relatively recent identification of T cell subsets has allowed the categorization of type IV hypersensitivity reactions into four subtypes, based on the type of cells involved, pathogenesis, and cytokine profile [21]. Type IVa is a reaction involving type 1 helper T cells (Th1), which activate and stimulate macrophages to produce such cytokines as interferon γ (IFN-γ) and TNF-α (e.g., contact dermatitis) (Figure 4A). In type IVb reactions, Th2 cells produce IL-4, IL-5, and IL-13, which initiate the production of IgE by B lymphocytes and the inactivation of macrophages (e.g., DRESS syndrome) [34]. Moreover, reactions of this type may also be involved in the late phase of allergic bronchitis or rhinitis (i.e., asthma and allergic rhinitis) (Figure 4B). IVc hypersensitivity is mainly mediated by Tc-released mediators, such as perforins, granulysin, and granzyme B to kill target cells (e.g., Stevens-Johnson syndrome) (Figure 4C). Type IVd reactions cause tissue damage due to the production of CXCL-8 (IL-8) by T lymphocytes, which promotes the recruitment of neutrophils to sites of inflammation (e.g., Behçet’s disease) (Figure 4D) [27,33,35]. 

## 4. Allergy Mechanism

The immune system responds to an allergen in two phases: early (sensitization) and late (effector) [39]. During the early phase, which may occur many years before the onset of clinical symptoms, sensitization to a specific allergen is initiated [26,40]. Allergens are recognized by major histocompatibility class II (MHC II) molecules on the surface of antigen-presenting cells (APCs). Antigen-MHC complexes are detected by Th lymphocytes as foreign, which results in the differentiation and activation of Th2 lymphocytes [41]. The activation of these cells leads to the production of inflammatory cytokines, such as interleukin 5 (essential for eosinophilic inflammation), interleukin 9 (stimulates mast cell proliferation), and interleukins 4 and 13, which stimulate B lymphocytes to switch the class of immunoglobulins to type E (IgE) [42]. Allergen-specific IgE antibodies bind to FcεRI receptors present on the surface of mast cells and basophils. Re-exposure to the allergen causes cross-linking of the IgE-FcεRI complexes, which in turn stimulates degranulation of mast cells and basophils, releasing inflammatory mediators responsible for development of specific allergic symptoms within a few minutes (Figure 5) [43,44]. Upon allergen activation, group 2 innate lymphoid cells (ILC2s) reside in mucosal tissues and secrete copious amounts of IL-5 and IL-13. IL-5 induces eosinophil recruitment, while IL-13 causes smooth muscle contraction and subepithelial fibrosis, resulting in tissue remodeling and airway hyperresponsiveness [45]. Mast cells and basophils deliver two types of mediators. Histamine, serotonin, and tryptase are constitutive mediators released by exocytosis. Other mediators, i.e., prostaglandins and leukotrienes, are synthesized de novo, and when released from the cell, they act as pro-inflammatory signaling molecules [46]. Recruitment of inflammatory cells, including eosinophils, basophils, and T cells, enhances the release of histamine and leukotrienes as well as other compounds, including pro-inflammatory cytokines and chemokines, sustaining the allergic response and promoting a late-phase reaction that may occur 6-9 h after primary exposure to an allergen [47,48].

## 5. Fungal Aeroallergens

Airborne fungal spores are now considered one of the leading causes of respiratory allergies [49,50,51]. Their concentration (230–106 spores/m^3^) exceeds the pollen concentration in the atmosphere by 100–1000 times [52]. The prevalence of respiratory allergy to fungi is not fully known. Still, it is estimated to affect about 20–30% of sensitive subjects (predisposed to allergies), or up to 6% of the general population [49]. The list of fungal allergens officially approved by the Subcommittee on Nomenclature of the International Union of Immunological Societies (IUIS) includes 105 isoallergens and variants from 25 fungal species belonging to the Ascomycota and Basidiomycota [51]. However, the number of fungal proteins capable of causing type I hypersensitivity reactions described in the literature is much higher, even if many of these allergens need to be better characterized. The catalog of defined fungal allergens [53] lists 174 allergens from the phylum Ascomycota and 30 from Basidiomycota. However, this list only includes a few fungal allergens partially characterized in their primary sequence [51]. Hypersensitivity to fungal spores is mainly induced by representatives of the genera *Alternaria*, *Cladosporium*, *Aspergillus*, *Penicillium*, and *Fusarium* [50].

The occurrence of fungal spores in the air is seasonal. Their peak concentration is recorded mainly in summer due to the availability of nutrients in the soil, favorable temperature, and humidity and in early autumn when rainy days are followed by sunny, dry, and windy days [54,55]. Many spores occur not only in the external environment but also indoors. Fungal spores present in the environment enter buildings with air or are carried by humans and animals. A high concentration of spores indoors is particularly detected in conditions of increased humidity, poor ventilation, or air-conditioning systems [55,56].

## 6. Clinical Manifestations of Fungal Hypersensitivity

Allergy to fungi manifests itself as immediate type I hypersensitivity mediated by immunoglobulin E, and as type II, III, and IV reactions. There is often an interaction between mechanisms in the pathogenesis of hypersensitivity reactions, which is reflected primarily in allergy to fungal spores [50,51]. The vast spectrum of clinical symptoms caused by fungal allergens includes rhinitis, conjunctivitis, urticaria, or atopic dermatitis [57]. In addition, the small size of the spores, usually not exceeding 10 µm (*Aspergillus fumigatus* 3.5–5.0 µm; *Aspergillus niger* 3.0–4.5 µm; *Cladosporium macrocarpum* 5–8 µm; *Penicillium brevicompactum* 7–17 µm), allows penetration into the lower respiratory tract, which in turn often leads to the development of such allergic reactions as asthma and allergic alveolitis [50,57].

### 6.1. Allergic Rhinitis (AR)

Many species of fungi induce AR, with *Alternaria*, *Aspergillus*, *Bipolaris*, *Cladosporium*, *Curvularia*, and *Penicillium* [52] as the most important factors. T helper lymphocytes are crucial in initiating the allergic immune response in AR through secretion of cytokines IL-5, IL-10, IL-13, and IL-4, which induces switching the classes of antibodies produced by B lymphocytes. Cross-linking allergens with their specific antibodies IgE on the surface of mast cells leads to rapid release of pre-formed inflammatory mediators, such as histamine, triggering early nasal allergic reaction symptoms, such as itching, sneezing, rhinitis, nasal congestion, and sometimes hyposmia, within minutes [21,58]. Histamine, TNF-α, and newly synthesized lipid mediators (leukotriene C4 and prostaglandin D2) contribute to the influx of inflammatory cells, i.e., eosinophils, basophils, and Th cells. The influx of these cells characterizes the onset of the late-phase allergic reaction, with nasal obstruction, hyposmia, and nasal hyperreactivity as the main symptoms [59]. The presence of allergen-specific IgE and eosinophilic rhinitis are typical features of AR, distinguishing it from other forms of this disorder [60,61]. Risk factors for AR include family predisposition, ethnic origin, high socioeconomic status, environmental pollution, and exposure to allergens or alcohol abuse [62,63]. Allergic rhinitis is classified as seasonal or annual, depending on sensitization to cyclic or year-round allergens, respectively [64,65].

In many cases, AR decreases the quality of life, which is manifested by poorer sleep quality, distraction, fatigue, irritability, and emotional disorders [59]. Much evidence points to a link between allergic rhinitis and asthma. Epidemiological studies have shown that these conditions often coexist in patients. At least 60% of patients with asthma suffer from rhinoconjunctivitis, while 20% to 30% of patients with allergic rhinitis have asthma symptoms [11,61]. Allergic rhinitis is the most critical risk factor for asthma and usually precedes its onset, which largely depends on the duration and severity of allergic rhinitis [66].

Moreover, allergic rhinitis enhances bronchial Th2-driven inflammation and the development of asthma [67]. There is also evidence of a link between sinus disease and allergic rhinitis. Some 25–30% of patients with acute sinusitis suffer from AR, as do 40–67% of subjects with unilateral chronic sinusitis, and up to 80% with bilateral chronic sinusitis [68]. The coexistence of AR and allergic conjunctivitis is equally shared, characterized by intense itching and hyperemia of the eye, lacrimation, and periorbital edema [69]. It occurs in about 50–70% of allergic rhinitis patients and is a symptom that best differentiates AR from other forms of this disease [59].

### 6.2. Allergic Fungal Rhinosinusitis (AFSR)

Allergy to fungi is associated with chronic rhinosinusitis (CRS) [51]. Allergic fungal rhinosinusitis (AFSR) is a distinct syndrome accounting for 5 to 10% of chronic rhinosinusitis [70]. Safirstein reported the first case of AFSR as a symptom of allergic bronchopulmonary aspergillosis (ABPA) in 1976 [71]. The diagnostic criteria for AFSR, in addition to the presence of chronic rhinosinusitis with nasal polyposis, include the presence of “allergic mucin” (also called “eosinophilic mucin”) [72,73]. It is a thick, viscous secretion characterized by substantial accumulation of eosinophils, often showing signs of degranulation (Charcot–Leyden crystals) and presephae [70,74]. The causality of AFSR is mainly attributed to *Aspergillus* species and fungi responsible for phaeohyphomycosis infection, including *Cladosporium*, *Bipolaris*, *Curvularia*, *Exserohilum*, and *Alternaria* [75,76].

### 6.3. Allergic Bronchopulmonary Mycosis (ABPM)

Most often, ABPM results from the spread of fungi in the respiratory tract [77]. An example of ABPM is allergic bronchopulmonary aspergillosis (ABPA) characterized by a severe hypersensitivity reaction to *Aspergillus fumigatus* antigens released during airway colonization [78]. Immunologically, ABPA is a mixed type I, type III, and type IV hypersensitivity caused by *A. fumigatus* colonization of the bronchi, with symptoms ranging from asthma exacerbation to severe, possibly fatal, lung damage [79]. Patients suffering from cystic fibrosis and asthma are particularly exposed to ABPA (ABPA occurs in about 7–22% of asthmatics) [77,79,80]. The symptoms of ABPA are primarily wheezing and pulmonary infiltrates, which may lead to pulmonary fibrosis and/or bronchitis [50].

Moreover, the exaggerated immune response induced by *A. fumigatus* is associated with a constellation of immune manifestations, including elevated serum IgE, severe pulmonary and peripheral blood eosinophilia, increased mucus production, and bronchiectasis [35,81]. Although it is speculated that *A. fumigatus* conidia and mycelium may remain in the airways long enough to release potent antigens that inhibit ciliary movement and/or initiate lung architecture remodeling, the exact mechanism of *A. fumigatus*-induced allergic respiratory disease is not known [82]. The pathophysiology of ABPA involves structural abnormalities in the airway epithelium that allow spores to break through the immune barrier and germinate into hyphae. During germination, spores secrete proteases that can disrupt the integrity of the epithelial barrier [83,84]. Several *A. fumigatus* allergens have been characterized [85,86], and studies of the Asp f1 allergen showed its strong homology with the mitogillin toxin (the consistency of protein sequences was equal to 95%) [87]. *Candida albicans*, *Curvularia*, *Geotrichum*, and *Helminthosporium* can also initiate the onset of symptoms characteristic of ABPA. However, their occurrence frequency is much lower than ABPA, and there are only a few documented cases [53,80].

### 6.4. Allergic Asthma

Asthma is characterized by the activation of mast cells and the release of mediators that initiate bronchial smooth muscle contraction, thereby leading to airway obstruction [88,89,90]. The ongoing inflammatory process is accompanied by damage to the airway epithelium, high levels of serum IgE, increased mucus production, influx and activation of eosinophils, and production of cytokines, mainly by Th2 lymphocytes [91]. Asthma manifests itself by wheezing, shortness of breath, chest tightness, coughing, and hyperreactivity [92]. During the asthmatic process, the airways develop inflammation involving a set of cells: eosinophils, lymphocytes, mast cells, and neutrophils. Epithelial cells, fibroblasts, myofibroblasts, and smooth muscle cells also play an essential role [93,94].

Asthmatic airway inflammation is characterized by infiltration of eosinophils, manifested by an increase in their number in the bronchoalveolar fluid, with a decreasing number of peripheral eosinophils. IL-13, histamine, prostaglandin type 2, and chemokines (RANTES, eotaxins, MCP-4) are responsible for the recruitment of eosinophils to the respiratory tract [95,96]. Neutrophils are mainly found in the airways of severe asthmatics [97]. These cells can release mediators, such as platelet-activating factor (PAF), thromboxanes, and leukotrienes, which initiate bronchial hyperreactivity and airway inflammation. Neutrophils secrete proteases and free radicals, thereby causing tissue damage [95]. Dendritic cells act as antigen-presenting cells and are stimulated directly or indirectly by allergens or by mediators (IL-25, IL-33, GM-CSF) of airway epithelial cells, respectively. It has also been found that dendritic cells can recruit eosinophils at the antigen presentation site and influence the differentiation of T lymphocytes [96,98].

Upon re-exposure to an allergen, mast cells become activated by cross-linking IgE Fc receptors on their surface, or by such stimuli as complement components C5a and C3a, releasing mediators responsible for bronchoconstriction and perpetuating the underlying inflammation. Mast cells are a source of histamine, cysteinyl leukotrienes, prostaglandins, cytokines, and PAF [95,99]. The inflammatory response in asthmatic airways is a complex interaction between the respiratory epithelium and the immune system. The quest for a chronic inflammatory response begins with the production of bioactive mediators from the airway epithelium that attract, activate, and recruit inflammatory cells to the lung airways. Infiltrated cells intensify the inflammatory response by releasing other biochemical mediators. The inflammatory mediators released by these cells are effectors of chronic inflammation, including (a) Th2 cytokines, (b) pro-inflammatory cytokines that promote and enhance the inflammatory response, (c) chemokines that are chemoattractants for leukocytes, and (d) growth factors [100,101]. Cytokines produced by Th2 cells, also called lymphokines, play an essential role in immunoregulation. They act on target cells, having a wide range of cellular functions such as activation, proliferation, chemotaxis, immunomodulation, releasing inflammatory mediators, cell growth and differentiation, and apoptosis. In contrast to acute and subacute inflammatory reactions, cytokines dominate in maintaining chronic inflammation [92,94,95,102].

IL-4, IL-5, and IL-13 are key pathophysiologic cytokines in asthma [96,103]. The primary cellular sources of IL-5 include Th2 lymphocytes, group 2 innate lymphoid cells (ILC2), mast cells, and eosinophils. Interleukin 5 is the main factor regulating the activation, proliferation, and maturation of eosinophils and promoting their migration from the bloodstream to the respiratory tract [104,105]. IL-4 and IL-13 are produced by various inflammatory cells, including activated Th2 cells, mast cells, basophils, and eosinophils. They are involved in differentiation of Th cells into Th2 cells, switching B cells to IgE production, airway remodeling, and mucus overproduction [90,103]. Another group of cytokines to be considered are pro-inflammatory cytokines, such as IL-1, TNF-α, IL-6, IL-11, granulocyte and macrophage colony-stimulating factor (GM-CSF), and stem cell factor (SCF). They may play a role in disease severity and resistance to anti-inflammatory therapy in asthma [94]. The pleiotropic activities of these cytokines include pro-inflammatory actions, such as leukocyte recruitment through increased expression of adhesion molecules on vascular endothelial cells, and induction of cytokine and chemokine synthesis [106]. GM-CSF is essential for the production and differentiation of macrophages and neutrophils and their survival in the airways [94,107]. Elevated levels of IL-1β, characteristic of severe asthma, are associated with macrophage activation and neutrophilic inflammation, similar to the role of TNF-α [108]. In addition, as shown by in vitro studies conducted in 1998, TNF-α plays a vital role in bronchial hyperresponsiveness and airway remodeling in asthmatics [109], which was confirmed by later experiments [110]. Lung alveolar macrophages and airway endothelial cells are the primary source of chemokines that directly contribute to the development of asthma. In asthma, chemokines that have a chemotactic effect on eosinophils deserve special attention [111]. The chemokines RANTES (CCL5), MCP-3 (CCL7), and MCP-4 (CCL113), which recruit eosinophils via CCR3 receptors, have been identified in the airways of asthmatics [111,112,113]. In addition, the CCR3 receptor can be activated by eotaxin-2 and eotaxin-3, resulting in eosinophil degranulation and release of damaging epithelial proteins [114,115].

Moreover, the same chemokines affect basophils and Th2 helper lymphocytes [112,116]. Growth factors are involved in the proliferation and differentiation of smooth muscle cells derived from various tissues, including the vascular system and the respiratory tract. They potentially contribute to an increase in the airway smooth muscle mass, which is observed in patients with chronic severe asthma, by stimulating the proliferation of airway smooth muscle cells. The different growth factors involved in the pathophysiology of asthma include PAF, transforming growth factor (TGF-β), nerve growth factor (NGF), fibroblast growth factor (FGF), epidermal growth factor (EGF), and insulin-like growth factor 1 (IGF-1) [109,117,118,119].

Dysfunction in the airway epithelial barrier plays a crucial role in sensitization to allergens and asthma pathogenesis [120,121]. Pattern recognition receptors (PRRs), Toll-like receptors (TLRs), retinoic acid-inducible gene-like 1 receptors (RIG-1), NOD-like receptors, C-type lectin receptors, protease-activated receptors (PARs), and purinergic receptors are expressed on airway epithelial cells recognizing pathogen-associated molecular patterns (PAMPs) and molecular pattern damage-associated proteins (DAMPs) released from dying or damaged cells [122,123]. This results in the release of pro-inflammatory cytokines/chemokines, such as IL-6, IL-8, CCL20, CCL17, TSLP, IL-25, IL-33, and GM-CSF, which can activate the effector cells of the innate and adaptive immune systems [124,125]. Exposure to allergens is associated with pathological structural changes in the epithelial barrier, manifested by the release of growth factors, e.g., EGF and TGF-β, which activate fibroblasts and myofibroblasts [118,126]. Its consequence is the excessive deposition of extracellular matrix components, leading to subepithelial fibrosis, thickening of the airway wall, and increased smooth muscle mass [127,128]. Vascular endothelial growth factor (VEGF) released by airway cells promotes angiogenesis and increases the size of airway wall vessels [129]. Features indicating a lack of integrity of the airway epithelial layer include detached ciliary cells, creoles in the sputum, and increased allergen permeability [130,131]. One of the critical features of epithelial remodeling in asthma is the loss of proteins responsible for mechanical cell–cell coupling, ensuring tight barrier integrity. The intercellular connections include apically located tight junctions, adjacent junctions, and basolaterally arranged (hemi)desmosomes [125]. E-cadherin is a protein specific for adherent junctions. Its extracellular domain connects neighboring cells, and the intracellular environment is linked with the actin elements of the cytoskeleton via α-, β-catenin, and p120 protein. E-cadherin is believed to be vital in forming other intercellular connections, and its disruption contributes to delocalization of proteins forming tight connections [132,133]. Proteins typical of tight junctions are also zonula occludens-1, i.e., ZO-1, occludins, and claudins; their role is to regulate epithelial permeability [127,134]. Disrupted expression of E-cadherin, β-catenin, ZO-1, and occluding, resulting in impaired barrier function, has been observed in the airway epithelium of asthmatics [130,131,135,136,137]. Moreover, loss of E-cadherin causes epithelial denudation with specific loss of ciliary cells and proliferation of goblet cells with inhibition of their differentiation and promotion of epithelial-mesenchymal transition (EMT) cells [104,138,139]. The consequences of the inability to restore the epithelial barrier’s function include increased allergen permeability, hyperreactivity, and remodeling of the airways, resulting from pro-inflammatory reactions and disrupted repair processes in the airways [109,125,126].

### 6.5. Severe Asthma with Fungal Sensitization (SAFS)

SAFS is characterized by severe asthma and sensitization to allergens of *Alternaria, Cladosporium*, *Candida albicans*, or *Aspergillus* fungi [140]. SAFS differs from ABPA in the absence of pulmonary infiltrates, bronchiectasis, and mucus retention in the airways. According to the WHO, the disease affects up to 33.9 million people worldwide, making it the most common respiratory disorder associated with *A. fumigatus* [141].

## 7. Impact of Climate Change on Fungal Aeroallergens

Climate change, mainly caused by increased concentrations of carbon dioxide and other greenhouse gases in the atmosphere, manifests in increased temperature and humidity and changes in the amount and distribution of atmospheric precipitation [142,143]. Extreme weather events, such as heat waves, heavy rainfall, and storms, will increase over the next few decades. These climate-related factors can affect the physiology and distribution of living organisms, such as plants and fungi. In this context, there is evidence that climate change has an impact on pollen and spore production by plants and fungi and on various phenological events [144,145]. This is reflected in the production, distribution, dispersion, and content of aeroallergens in the air, which may result in changes in the incidence of allergic diseases and/or the severity of their symptoms [146]. Increases in temperature change the phenology of many living organisms, including fungi. The reaction of airborne spores to temperature changes is difficult to predict because their concentration in the air is affected by a set of meteorological parameters [147]. Studies conducted so far indicate an extension of the fruiting seasons of allergenic fungi, which is the cumulative effect of higher air temperature and lower rainfall [140,148]. Climate change has also been found to induce morphological changes in fungal spores. Kauserud et al. found that spores produced at the beginning of autumn exhibited higher water accumulation, which increased their size, while spores produced at the end of autumn were smaller [145]. In terms of allergy, changes in the spore size are important because a smaller size makes spores more accessible and inhalable; hence, they are more likely to be deposited deeper in the human respiratory system. In addition, spore enlargement was observed in early autumn, during a period of elevated average air temperature and lower rainfall [140].

Rapid weather changes, such as floods, storms, and hurricanes, can disperse fungi, bringing very rare or unknown fungal species into new areas. Additionally, extreme weather events can increase the number of spores in the air. For instance, in the aftermath of Hurricane Katrina in New Orleans, USA, high indoor and outdoor fungal counts were noted [149,150]. Thunderstorms and the occurrence of asthma are correlated with a doubling of fungal spores in the environment [147,151]. There have been many documented cases of asthma attacks during thunderstorms, not only in asthmatics but also in subjects who previously only suffered from allergic rhinitis. So-called “thunderstorm asthma” is characterized by the onset of asthmatic symptoms possibly caused by the extensive dispersion of inhaled allergenic spore particles due to osmotic rupture [152,153]. Since the phenomenon was first reported in the UK in 1985 [144], several parts of the world have experienced successive episodes characterized by increased emergency room visits and hospitalizations that correlate with thunderstorm seasons and high airborne spore concentrations [154]. According to current climate change scenarios, the intensity and frequency of heavy rainfall episodes, including thunderstorms, will increase over the next several decades, which can be expected to be associated with an increase in the number and severity of asthma attacks [155,156]. Climate change also leads to changes in seasonality (e.g., later winters or earlier springs). This may affect the appearance of fungi at different times than before, leading to an increase in fungal allergens in the air in new seasons. In the southern Indian region, Priyamvada et al. found a large seasonal variation in the occurrence of three allergenic fungi: *Cladosporium cladosporioides*, *Aspergillus fumigatus*, and *Alternaria alternata* [157]. A change in air quality can affect fungal allergens, as fungi growing in polluted air can produce more allergens. In addition, particulate matter (PM) can interact with airborne allergens, such as fungal spores, increasing the risk of sensitization and worsening asthma and hay fever symptoms [158,159,160].

### Fungal Plant Parasites as a Source of New Allergens—Impact of Climate Change

Humans have transported various plants to new lands, for food, medicine, or ornamental purposes, for centuries. Some species find a new niche in which they can grow, develop, and even become dominant. Sometimes, such invasive plant species harm their new ecosystem. Their fungal parasites may also have a harmful effect. An example is the accidental import of *Cryphonectria parasitica* from Asia, which has destroyed many tall chestnut trees in forests on the east coast of the United States [161].

As a result of climate change, plants can also inhabit new areas and introduce fungal spores, exposing humans to contact with as yet unencountered novel allergens. Although *Alternaria*, *Aspergillus*, *Cladosporium*, *Penicillium*, and *Fusarium* are the most critical allergenic fungi, it cannot be excluded that very common native and invasive phytopathogenic microfungi causing mass plant infestations are also a source of allergens. Phytopathogenic microfungi are plant parasites commonly found in the human environment [162,163]. They cause mass plant infestation and are responsible for reducing yields and deteriorating the quality of plant products and the decorative value of ornamental plants. The physiology and distribution of plants and fungi, as well as pollen and spore production, depends on geographical location, air quality, human activity, and the local source of vegetation [144,151]. Besides native phytopathogenic microfungi, over the last few years, massive plant infestation of invasive microfungi has been observed, e.g., in Poland, mainly from Asia, North America, and even Australia. These microfungi primarily belong to the *Erysiphales* and *Puccinales* orders [164,165,166,167].

The arguments for the high probability of induction of allergic reactions by phytopathogenic fungi are as follows:−plant parasites are commonly found in the human environment;−large numbers of spores are produced on the surface of infected plant organs and in the air. In favorable environmental conditions, spores are released into the air in enormous numbers. Estimates indicate that the number of fungal spores on the surface of infected plant organs and in the air is comparable to the amount of vascular plant pollen [168];−microfungal spores and fruiting bodies are microscopic, readily carried by air currents, and can be inhaled into the human respiratory tract. The dimensions of conidia, urediniospores and teliospores of microfungi are within the limits of 16–63 × 12–25 µm [169,170]. The fruiting bodies are more extensive, with a diameter of 105–270 µm. However, such sizes do not exclude their penetration into the respiratory tract of humans and animals;−the presence of chitin in the cell wall (in species from the kingdom Fungi) can induce severe allergic reactions [50];−representatives of the kingdom Chromista can cause an allergic response in humans [171]. Our preliminary studies have indicated that *Peronospora lamii* produces large amounts of arachidonic acid, which is responsible for development of human inflammation (unpublished data).

Although many existing and potentially invasive plant species spread into new areas as stowaways and on cargo ships, they must find favorable conditions for settlement. Thus, climate change may favor the colonization of new regions by invasive plant species, along with their fungal parasites. This may be associated with introduction of new allergens to these territories [149,172].

## 8. Conclusions

Fungi are essential but still underestimated sources of allergens. The increasing incidence of allergic respiratory diseases suggests the need for extension of diagnostics to include new species of fungi. Phytopathogenic microfungi that parasitize many common crops, ornamental plants, and weeds may be such unique potential allergens. In addition, climate change may contribute to expanding the range of many plants and their fungal pathogens.

## Figures and Tables

**Figure 1 jof-09-00544-f001:**
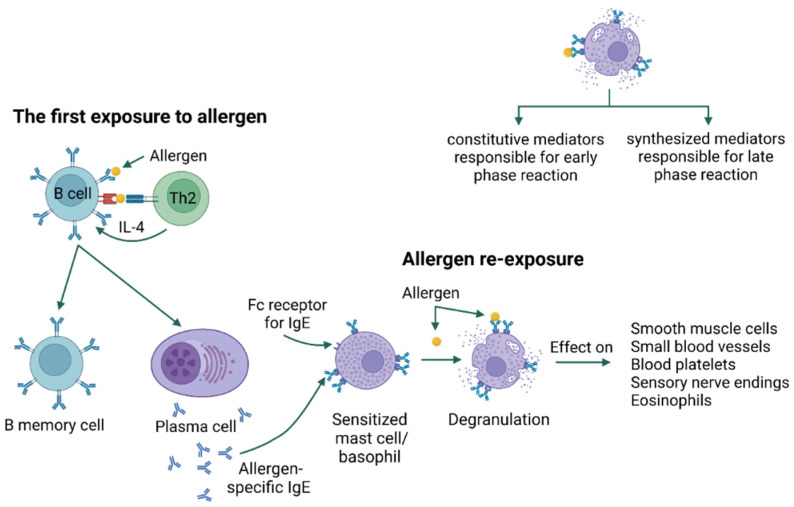
Type I hypersensitivity [25,26] [created with BioRender.com accessed on 8 April 2023].

**Figure 2 jof-09-00544-f002:**
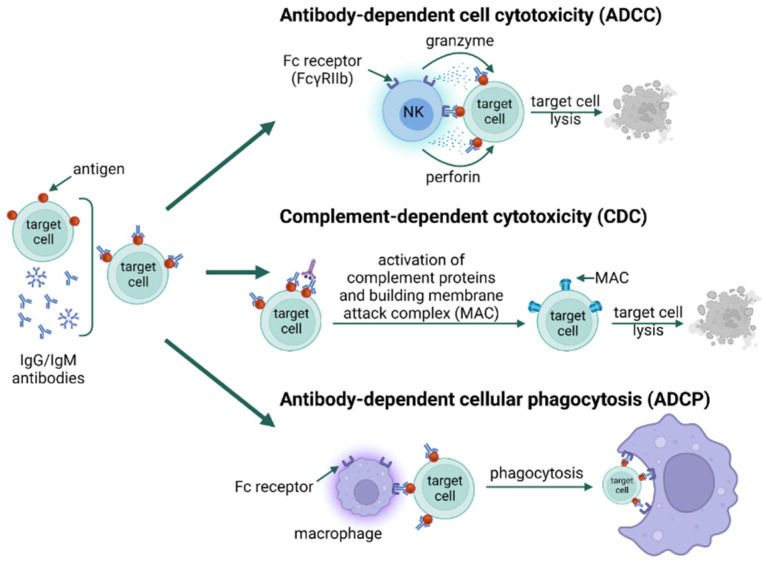
Type II hypersensitivity; type IIa ADCC, type IIa CDC, type IIa ADCP [25,29,30] [created with BioRender.com accessed on 8 April 2023].

**Figure 3 jof-09-00544-f003:**
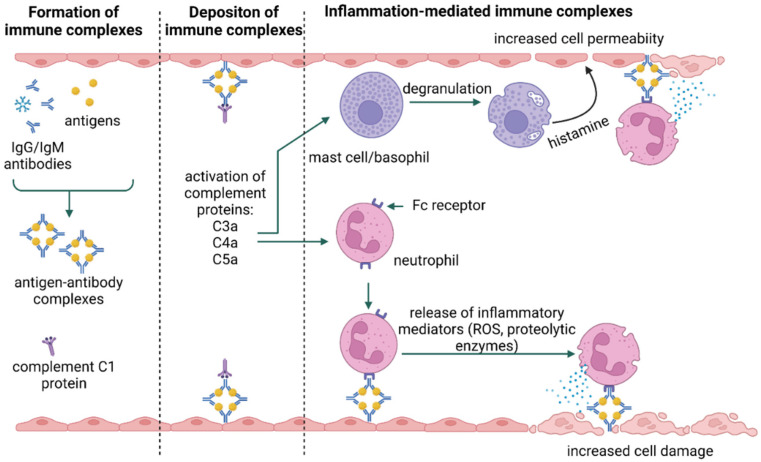
Type III hypersensitivity [25,32] [created with BioRender.com accessed on 8 April 2023].

**Figure 4 jof-09-00544-f004:**
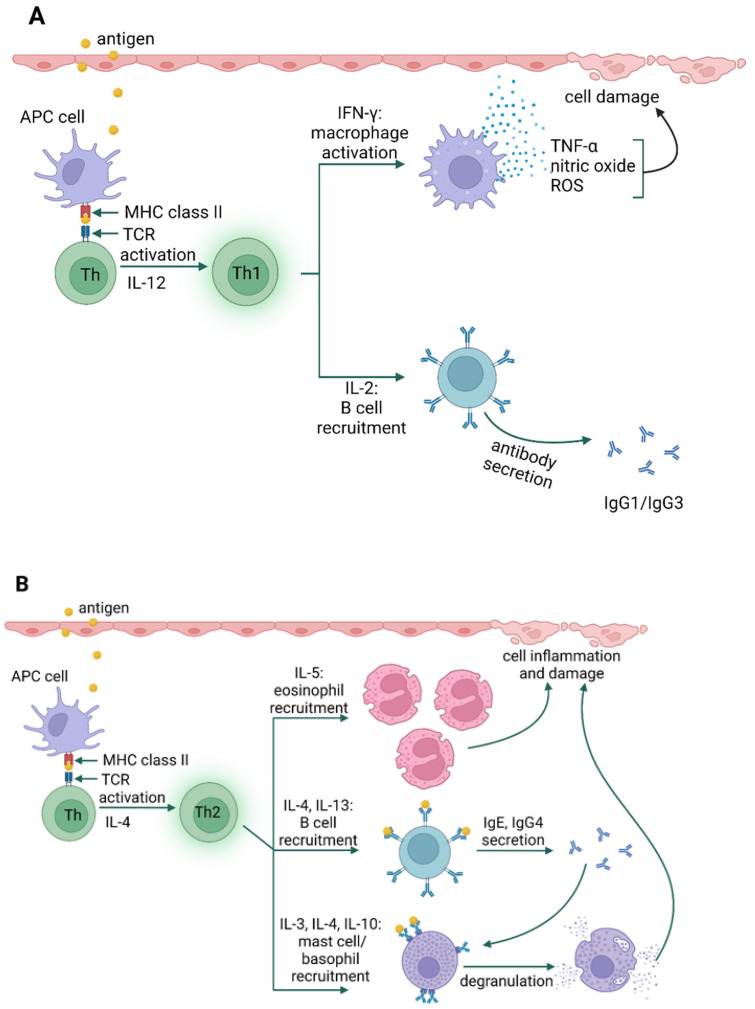
Type IV hypersensitivity; type IVa (**A**), type IVb (**B**), type IVc (**C**), type IVd (**D**) [35,36,37,38] [created with BioRender.com accessed on 8 April 2023].

**Figure 5 jof-09-00544-f005:**
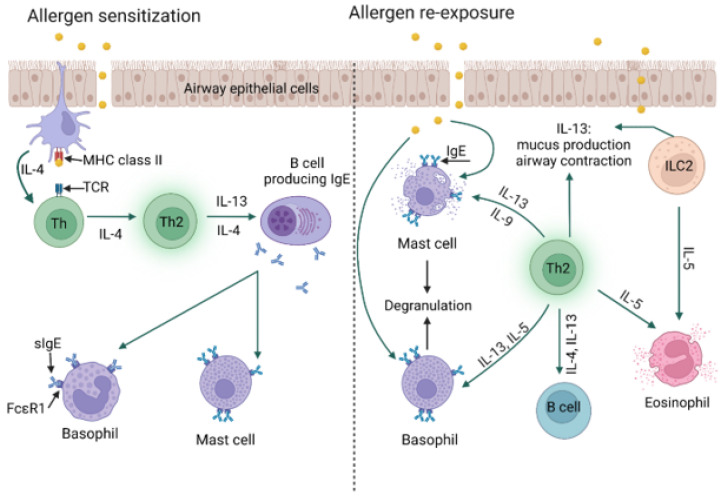
Allergy mechanism [25,40,41,43,44] [created with BioRender.com accessed on 8 April 2023].

## Data Availability

No new data were created in the manuscript.

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
