# Peer review of "Fungal Aeroallergens—The Impact of Climate Change"

_jof, 2023, doi:10.3390/jof9050544_

Round 1

Reviewer 1 Report

In this review, authors address an underestimated field to discuss the impact of environmental fungal spores on allergic diseases and their association with human activities and climate change.  The references cited in the review support the characterization of epidemiology, effect mechanisms of allergy and hypothesis in the end of review, which is that phytopathogenic microfungi could be a new classes of fungal allergens. 

Review was well written and I accept it for publication after clarify a few typing errors.  

Page1, line 16, on the development the allergic disease,  word "of" may need to be inserted after "development".

Page 8, line 208-209, all the species names need to be italicized. 

Page 13, line 482-486, this sounds a orphan paragraph. If you cited others, please mark the reference and reduce the font size of this para.

Page 14, line 496-497, the font of this paragraph seems bigger than other subtitles, and MST is bolded that is different from ASC.

Author Response

Thank you kindly for reviewing our paper and helpful comments. Below, please find the response to other comments.

Response to Reviewer 1 Comments

Reviewer 1: In this review, authors address an underestimated field to discuss the impact of environmental fungal spores on allergic diseases and their association with human activities and climate change.  The references cited in the review support the characterization of epidemiology, effect mechanisms of allergy and hypothesis in the end of review, which is that phytopathogenic microfungi could be a new classes of fungal allergens. 

Review was well written and I accept it for publication after clarify a few typing errors.  

Authors: Thank you kindly for reviewing our paper and helpful comments. Below, please find the response to other comments.

Point 1: Page1, line 16, on the development the allergic disease,  word "of" may need to be inserted after "development".

Response 1: We corrected this sentence: „After presenting the basic information on the mechanisms of allergic reactions, we describe the impact of fungal allergens on the development of the allergic diseases”.

Point 2. Page 8, line 208-209, all the species names need to be italicized. 

Response 2: We corrected all the species names into italic: „Many species of fungi induce AR, with Alternaria, Aspergillus, Bipolaris, Cladosporium, Curvularia, and Penicillium [52] as the most important factors”.

Point 3: Page 13, line 482-486, this sounds a orphan paragraph. If you cited others, please mark the reference and reduce the font size of this para.

Response 3: We formatted this paragraph and added references No 149 and 172 (172 is the new one).

Point 4: Page 14, line 496-497, the font of this paragraph seems bigger than other subtitles, and MST is bolded that is different from ASC.

Response 4: We formatted this part according to reviewer’s suggestion.

Additionally, we reintroduced Figure 4A to avoid word cutting.

Reviewer 2 Report

This article is generally well written and adds an important consideration to the underappreciated area of allergic disease and its continuing growth.  I have no major concerns but some minor comments below:

1.  General comment on references;  I have not checked every reference but would recommend a thorough review by the authors as some seem mis-placed for example, 

Section 5.3, line 260, prevalence of ABPA - the generally accepted source for this is Denning (reference 79) but this is not referenced

Section 5.5, line 393, reference 141 refers to changes in fungal fruiting patterns - I believe there are better references for SAFS prevalence

Section 6 - line 411, reference is the date rather than the reference number (145)

2. Formatting - there are several places where the text format changes to italics - Section 6, line 417 and Section 6.1, line 482 for example.  

3. Language - although the English is excellent, some further review of language would be of benefit.  Examples include

Lines 31/32 - description of rhinitis and asthma seem co-mingled and a little confusing

Lines 43/44 - language is slightly too dramatic referencing patients in 'fear' - should focus on the specifics of the prevalence.

Lines 57/58 - language is confusing and unclear - are the authors saying that 50% of adults over 30 who are newly diagnosed have an allergic phenotype?

The English is of a high quality with only minor issues referenced in the review.  The manuscript would benefit from some additional review for language.

Author Response

Thank you kindly for reviewing our paper and helpful comments. Below, please find the response to other comments.

Response to Reviewer 2 Comments

Reviewer 2: This article is generally well written and adds an important consideration to the underappreciated area of allergic disease and its continuing growth.  I have no major concerns but some minor comments below.

Authors: Thank you kindly for reviewing our paper and helpful comments. Below, please find the response to other comments.

Point 1.  General comment on references;  I have not checked every reference but would recommend a thorough review by the authors as some seem mis-placed for example, 

Response: We thoroughly checked all references and their citations.

Section 5.3, line 260, prevalence of ABPA - the generally accepted source for this is Denning (reference 79) but this is not referenced

Response: We introduced reference No 79 to support the prevalence of ABPA.

Section 5.5, line 393, reference 141 refers to changes in fungal fruiting patterns - I believe there are better references for SAFS prevalence

Response: We introduced a more recent paper describing SAFS prevalence: Prasad R, Kazmi SA, Kacker R, Gupta N. Severe asthma with fungal sensitization. Indian J Allergy Asthma Immunol 2021, 35, 3-7. Therefore we changed the earlier sentence into a more appropriate one: „According to the WHO, the disease affects up to 33.9 million people worldwide, making it the most common respiratory disorder associated with A. fumigatus.

Section 6 - line 411, reference is the date rather than the reference number (145)

Response: We removed the date and introduced proper citations (145).

Point 2:. Formatting - there are several places where the text format changes to italics - Section 6, line 417 and Section 6.1, line 482 for example.  

Response 2: We carefully check the text to introduce proper formatting.

Point 3: Language - although the English is excellent, some further review of language would be of benefit.  Examples include

Lines 31/32 - description of rhinitis and asthma seem co-mingled and a little confusing

Response; We agree that such description is confusing, therefore, to clarify, we changed it into: „respiratory system allergies, i.e., allergic rhinitis with conjunctivitis and asthma with wheezing, coughing, and shortness of breath”.

Lines 43/44 - language is slightly too dramatic referencing patients in 'fear' - should focus on the specifics of the prevalence.

Response: The language is really dramatic – we agree. Therefore, we omitted such an interpretation: „Currently, more than 150 million Europeans suffer from chronic allergic diseases, and 20% struggle with severe and debilitating forms”.

Lines 57/58 - language is confusing and unclear - are the authors saying that 50% of adults over 30 who are newly diagnosed have an allergic phenotype?

Response: We agree with this Reviewer’s comment. Therefore, we clarify this part: „The most common asthma phenotype is allergic asthma. It is estimated that up to 80% of childhood asthma and more than 50% of adult asthma cases may have an allergic component”.

Additionally, we reintroduced Figure 4A to avoid word cutting.